# Targeting Wnt/β-Catenin Pathways in Primary Liver Tumours: From Microenvironment Signaling to Therapeutic Agents

**DOI:** 10.3390/cancers14081912

**Published:** 2022-04-10

**Authors:** Federico Selvaggi, Teresa Catalano, Roberto Cotellese, Gitana Maria Aceto

**Affiliations:** 1Unit of General Surgery, Ospedale Floraspe Renzetti, 66034 Lanciano, Chieti, Italy; fedeselvaggi@hotmail.com; 2Department of Clinical and Experimental Medicine, University of Messina, Via Consolare Valeria, 98125 Messina, Italy; tcatalano@unime.it; 3Department of Medical, Oral and Biotechnological Sciences, University “G. d’Annunzio” Chieti-Pescara, Via dei Vestini 31, 66100 Chieti, Italy; roberto.cotellese@unich.it; 4Villa Serena Foundation for Research, 65013 Città Sant’Angelo, Pescara, Italy

**Keywords:** liver, Wnt/β-catenin signaling, microenvironment, HCC, CCA, HB, surgical treatments, chemotherapy

## Abstract

**Simple Summary:**

Human primary liver cancers have a poor prognosis. At present, there are limited treatment alternatives, and to understand the molecular aspects of liver tumorigenesis it is essential to develop more efficient pharmaceutic modalities. Liver tumors have mutations in genes encoding key components of Wnt/β-catenin signaling, and two different molecular pathways have been identified, termed non-canonical and canonical networks. In this review, we discuss the dysregulation of the Wnt/β-catenin signaling network, especially in hepatocellular carcinoma, in cholangiocarcinoma and in patients with hepatoblastoma. Based on the current literature we propose an update of molecular mechanisms focusing on Wnt/β-catenin pathways in physiological conditions and in human primary liver carcinomas, together with an overview of preclinical and clinical studies. The clinical implication of Wnt inhibitors in association with conventional surgical and lo-co-regional therapies is reviewed.

**Abstract:**

Primary liver cancers (PLCs) are steadily increasing in incidence and mortality in the world. They have a poor prognosis due to their silent nature, late discovery and resistance to common chemotherapy. At present, there are limited treatment alternatives, and the understanding of PLC molecular aspects is essential to develop more efficient drugs and therapeutic surgical and loco-regional strategies. A clear causal link with liver damage, inflammation, and regeneration has been found in the occurrence of PLC over the last few decades. Physiologically, Wingless/It (Wnt)-β-catenin signaling plays a key role in liver development, metabolic zonation and regeneration. Loss of functional homeostasis of this pathway appears to be a major driver of carcinogenesis in the liver parenchyma. In the hepatic microenvironment, molecular deregulations that exceed the Wnt signaling biological capacity can induce tumor initiation and progression. Indeed, somatic mutations are identified in key components of canonical and non-canonical Wnt signaling and in PLCs and precancerous lesions. In this review, the altered functions of Wnt/β-catenin signaling are considered in human PLCs, with emphasis on hepatocellular carcinomas (HCC), cholangiocarcinomas (CCA) and hepatoblastomas (HB). Based on recent literature, we also focused on liver cancerogenesis through Wnt deregulation. An overview of preclinical and clinical studies on approved and experimental drugs, targeting the Wnt/β-catenin cascade in PLCs, is proposed. In addition, the clinical implication of molecule inhibitors that have been shown to possess activity against the Wnt pathway in association with conventional surgical and loco-regional therapies are reviewed.

## 1. Introduction

Primary liver cancer (PLC) is often diagnosed at an advanced stage. Worldwide, it is the fifth most common cancer, represents the third leading cause of cancer-related death and develops at a rate of 3% to 4% in patients with liver disease [1,2]. Despite efforts in basic and clinical research, the prognosis of patients with advanced liver cancer is often poor, and the 5-year survival rate is less than 5%. PLC in adults is composed of 90% hepatocellular carcinoma (HCC) and 10% cholangiocarcinoma (CCA) [3,4,5]. HCC is the third most common cause of cancer-related deaths worldwide [6,7,8]. Hepatoblastoma (HB) is a rare neoplasia, but it represents the most common malignant liver tumor in children [9].

Risk factors for HCC and CCA are heterogeneous and linked to environmental, infectious, and dietary lifestyle origins [1,10]. Exposure of liver tissue to repeated damaging events that exceed its adaptive biological capacity may result in chronic hepatitis, alcoholic cirrhosis, or nonalcoholic fatty liver disease (NAFLD) and metabolic dysfunction-associated fatty liver disease (MAFLD) [11], which ultimately predispose to HCC onset [1,12]. Although the incidence of HCC from simple NAFLD is less common than that reported for chronic hepatitis B or C from viral agents [12], HCC typically occurs in the context of chronic liver disease and cirrhosis. Hepatitis C virus (HCV), hepatitis B virus (HBV), and hereditary hemochromatosis can lead directly to HCC, whereas HCC related to other underlying liver diseases is linked to the development of cirrhosis [12].

Several risk factors, such as liver parasites, chronic biliary tract and liver diseases, including primary sclerosing cholangitis, choledochal cysts, Caroli’s disease and HCV cirrhosis, as well as the same aspects of lifestyle that cause chronic inflammation and cholestasis in the liver, have been linked to the development of CCA [3,4,13,14]. CCA develops along the biliary tree and is classified into intrahepatic, perihilar or distal extrahepatic tumours [15]. The incidence of CCA is variable in different geographical regions as a result of environmental and genetic conditions. The disease occurs after the fourth decade of life and more frequently in men than in women [14].

HB is a rare malignant liver cancer occurring in infants and children at the age of 2–3 years for about 0.5–2% of all pediatric tumors; its etiology remains largely unknown [16,17,18]. HB differs from HCC by the absence of underlying liver disease or viral infection [19]. The term “hepatoblastoma” suggests that these tumors may originate from hepatoblasts, or embryonic liver progenitor cells [18]. Most HBs are sporadic and are associated with extreme prematurity, younger maternal age, maternal smoking, higher maternal prepregnancy body mass index, low birth weight and congenital anomalies [19,20]. HBs are classified as epithelial, mixed epithelial and mesenchymal subtypes based on histopathological features. Moreover, HBs are classified as fetal type by presenting irregular hepatocytes as in fetal liver, as embryonal type by showing solid patterns with ribbons, rosettes and papillary formation, and finally a mixed-type of HB with stromal components chondroid and osteoid tissues [19].

Surgical resection is the treatment of choice for patients with well-preserved liver function. Liver transplantation is the option for patients who have underlying cirrhosis or in HB. Prevention is important for parasite-endemic and HBV-endemic regions, as well as health education, vaccination, antiviral therapies, and avoidance of toxin exposure [13]. Standard surveillance protocols should be developed for patients with biliary disorders, and cholelithiasis.

A clear causal link with liver damage, inflammation, and regeneration has been found over the last few decades in the occurrence of PLC [21]. Receiving both portal vein and arterial blood, the liver is regarded as a first-line immunological organ in the defense against blood-borne infections. This physiological role must also maintain the balance between immune activation and tolerance in host defense to avoid inappropriate immune responses against non-pathogenic exogenous molecules such as those derived from food [22]. Recent studies have also revealed the involvement of the gut-liver axis in the pathophysiological mechanism responsible for HCC [23]. Indeed, the intestinal barrier is essential to preserve the normal physiology of the host against microorganisms residing in the intestinal ecosystem (microbiota). If this barrier is disrupted, and/or in the presence of dysbiosis, bacterial endotoxins (e.g., LPS) can promote the onset and progression of liver disease. Therefore, increased bacterial translocation has been shown to be a key sign of HCC in animal models, whereas in humans, clinical studies in this regard are still limited [23]. During infections, pro-inflammatory and anti-inflammatory immune responses are regulated in a coordinated manner to protect the host and limit the pathological insults [24]. Significant changes in tissue metabolism occur during inflammatory and immune responses. These modifications include local depletion of nutrients, increased oxygen consumption, and generation of large amounts of reactive nitrogen and oxygen intermediates. Moreover, pathogens are often evolutionarily adapted to use host metabolic pathways and survival mechanisms for their own persistence and proliferation [24].

The Wingless/It (Wnt)/β-catenin signaling plays a critical role in liver development, metabolic zonation and regeneration [7,25,26]. Molecular dysregulations of this pathway are closely related to the liver microenvironment and promote development and progression of PLC. Liver tumors show mutations in genes encoding key elements of Wnt-β-catenin signaling. Mutations in the *CTNNB1* gene encoding β-catenin or in components of the degradation complex, such as *AXIN* and *APC* (encoding Axin and Adenomatous Polyposis Coli, respectively) have been detected in PLC [18,27,28,29]. In addition, two different pathways of Wnt signaling, termed canonical and non-canonical, have been identified. In adult liver, β-catenin and its negative regulator APC are localized in the perivenous and periportal areas to control liver metabolic activity [19]. Moreover, Wnt signaling is often aberrantly activated by oncogenic viruses and bacterial toxins to promote cellular transformation and carcinogenesis in target cells [30,31]. Oncogenic viruses have devised means to down-regulate inhibitory and/or up-regulate stimulatory proteins of Wnt signaling networks. Infectious agents involved in primary liver cancers such as HBV, HCV, and Epstein Barr virus (EBV), can modulate components of the Wnt pathway to subvert normal cellular processes including proliferation, differentiation, and cell survival [30].

Antitumor therapeutic perspectives aim to guide clinical practice in the direction of personalized treatments. The inhibition of Wnt-ligand production should be considered as a therapy in biliary diseases and primary liver cancers [32]. To this end, application research on novel molecular agonists and antagonists of Wnt-dependent signals aims to identify antisecretory therapies to which subsets of tumor tissues might selectively respond [33]. However, safe application of these novel drugs will require limiting the possibility of side effects on stem cell niches, which are necessary to maintain physiological homeostasis in the entire organism.

In this review, the functions of Wnt/β-catenin signaling are considered in light of its physiological and pathological roles in human liver. The molecular aspects of Wnt/β-catenin cascade in liver development and new insights on Wnt/β-catenin deregulation in PLCs are explored. Although more recent studies have underlined the specific genetic alterations and the oncogenic roles of aberrant Wnt/β-catenin signaling in liver cancer, especially in HCC patients as reported by current literature, we discuss the clinical implication of approved and experimental drugs that have been shown to possess activity against Wnt pathway in association with conventional surgical and loco-regional therapies.

## 2. Physiological Role of Wnt Signaling in Liver

The Wnt/β-catenin signaling controls a wide range of biological processes both during embryogenesis and in adult life. Historically, the Wnt pathway has been identified simultaneously as a guide for physiological processes of development and intercellular communication and in pathological processes such as in cancer [34]. This signaling involves a multitude of components, including ligands, receptors, co-receptors, agonists, and antagonists, that act in different autocrine, paracrine, and endocrine modes, thus regulating numerous cellular processes. In humans, 19 Wnt ligands and 10 different Frizzled (Fz) receptors are known to date. Wnt ligands are proteins of about 40 kDa that present many conserved cysteine residues [35,36]. Modifications in lipids and sugars determine their competence to interact with receptors in appropriate domains of the plasma membrane, so as to control their activity (see Figure 1). Wnt signals are transduced into cells through the canonical (β-catenin-dependent) and non-canonical (β-catenin-independent) pathways. Some Wnt ligands preferentially trigger the canonical pathway, while others stimulate the non-canonical pathway [37,38].

### 2.1. Wnt/β-Catenin-Dependent Signaling

Canonical Wnt/β-catenin-dependent signaling is triggered by one of the secreted Wnts by engagement of a receptor complex comprising a G-protein-coupled Frizzled family receptors (Fz) and low-density lipoprotein receptor-related protein 6 or 5 (LRP 5/6) that sends the biological signal to the Dishevelled protein within the cell [39,40,41] (Figure 1). The Wnt signal received by the receptor complex is then transduced into intracellular molecular pathways in a cascade process that addresses the needs of cell growth and differentiation [42]. In the absence of Wnt ligands, the cytoplasmic destruction complex formed by Adenomatous Polyposis Coli and Axin (APC/Axin) regulates the output of the canonical Wnt pathway by controlling the stability of β-catenin in the cytoplasm, where it is phosphorylated by two constitutively active serine-threonine kinases (CK1a/GSK3a/b). Phosphorylation of β-catenin prevents it from reaching the nucleus and binding to DNA-bound T cell factor/lymphoid enhancer factor (TCF/LEF) proteins, thereby repressing transcription of target genes [43] (Figure 1a). In addition to being an effector of the Wnt pathway, β-catenin is also a key player in cadherin-mediated cell adhesion. In the absence of Wnt signaling, β-catenin associates with E-cadherin at cell-cell junctions. This sequestration at the membrane limits its translocation to the nucleus, but also its degradation mediated by the destruction complex. Activation of the Wnt receptor inhibits the degradation of newly synthesized β-catenin and this allows a cadherin-free form of *de novo* synthesized β-catenin to enter the nucleus. In this way, β-catenin may also drive cancer cell survival by enhancing growth factor receptor signaling, such as Epidermal Growth Factor Receptor (EGFR) [44]. Mutations in molecules that are part of the Wnt/β-catenin pathway (particularly truncating mutations in the *APC* gene) lead to the formation of constitutive nuclear TCF/β-catenin complexes and uncontrolled transcription of target genes.

### 2.2. Wnt/β-Catenin-Independent Signaling

Non-canonical signaling (β-catenin-independent) can be further divided into the Wnt/Planar Cell Polarity (PCP) pathway and the Wnt/Ca^2+^ pathway. Both are activated by the binding of a Wnt ligand to Fz proteins and their co-receptor tyrosine kinase-like orphan receptor 2 (ROR2) to activate Dishevelled (Dvl) (Figure 2). Non-canonical Wnt pathways control the activity of small GTPases that alter actin machinery and cytoskeletal rearrangement. Thus, Wnt molecules can stimulate c-Jun N-terminal kinase (JNK) activity or promote cell adhesion and movement through calcium by activating Ca^2+^/calmodulin-dependent protein kinase II (CaMKII) and Calmodulin, that allows the transcription factor Nuclear factor of activated T-cells (NFAT) to translocate into the nucleus. NFAT, together with other cofactors (derived from other signal cascades) binds DNA. This induces the transcription of several genes including those for T lymphocyte activation (i.e., IL2) [45]. Dysfunctional alterations of these signals may therefore block the activation of lymphocyte cells and expose to the development of infectious diseases. Recent evidence indicates that non-canonical pathways of Wnt signaling are involved in the inflammatory response and subsequent oxidative stress [46,47,48,49,50].

### 2.3. Wnt/β-Catenin Signaling in Liver

The canonical Wnt/β-catenin pathway is integrated into evolutionarily conserved signaling processes during human embryogenesis. Gene expression control by Wnt/β-catenin plays a pivotal role in organogenesis driving both polarity and specific cell fate. The tight regulation of embryogenesis by the Wnt/β-catenin signal is also manifested in hepatobiliary development, hepatic maturation and zoning [28]. Liver and biliary tract are formed from a diverticulum that originates from the ventral part of the endoderm of the anterior intestine near its junction with the yolk sac and the middle intestine. The diverticulum develops within the mesoderm and then divides into a Y cranial and one ventral branch. From the first, the hepatic tissue is formed, from the second the gallbladder and the cystic duct, while the common tract constitutes the choledochus [51,52]. As well as for the development and maintenance of hepatobiliary functions, their ability to regenerate is also fundamental via Wnt-β-catenin. Residing cells within the liver bud are bipotent hepatoblasts, which rapidly proliferate, resulting in liver bud expansion, while remain undifferentiated. Hepatoblasts produce albumin and α-fetoprotein (AFP), both indicators of hepatic fate commitment [52]. Furthermore, Wnt appears to drive liver growth, through hematopoietically-expressed homeobox protein (Hhex) gene expression, hepatoblast proliferation and hepatocyte specification [53].

In vitro hepatic differentiation protocols demonstrate progressive loss of expression of the pluripotent marker Octamer-binding transcription factor 4 (OCT4) and increased expression of early-stage hepatic differentiation genes such as SRY-box transcription factor 17 (SOX17), GATA binding protein 4 (GATA4) and Forkhead box protein A2 (FOXA2) [54].

In the adult liver, β-catenin is localized in the perivenous area, while its negative regulator APC is in the periportal area. This dislocation allows them to exert control over hepatic metabolic activity [19,55]. During the later steps of liver development, differentiation of hepatoblasts generates two different liver cells: hepatocytes and cholangiocytes, or biliary epithelial cells (BECs). Hepatocytes represent the main parenchymal liver cell type and constitute 80% of liver mass [56]. Moreover, BEC proliferation is also specifically β-catenin dependent. Recent evidence in the study of metabolic zoning inside the hepatic lobule attributes specific connotations to hepatocytes, based on exclusive and location-dependent gene expression patterns [57].

The function of pericentral lobule hepatocytes appears to be regulated by modulation of the gene expression pattern in components of Wnt/β-catenin signaling. It is currently believed that the molecular basis of metabolic zoning in the liver directs both hepatocyte function and the limitation of initial hepatotoxic damage in specific areas. This allows for effective regeneration and restitution responses from unaffected cells. It has also been revealed that many pathological conditions in the hepatic lobule result from spatially deregulated mechanisms of hepatocyte adaptation and repair processes [57]. A pathological microenvironment may affect cell signaling and metabolic liver zonation driving liver diseases and carcinogenesis. Wnt signaling controls liver metabolism and proliferation through the expression of Hepatocyte Nuclear Factor 4 (HNF4) transcription factor-dependent genes. Thus, aberrant activation of β-catenin signaling is consistent with its involvement in liver cancer [58]. Oxygen tension also underlies the functional zonation of the liver. Indeed, the oxygenation gradient along hepatic sinusoids contributes to the determination of a critical microenvironment for specific hepatic functions, such as intermediate metabolism of amino acids, lipids, and carbohydrates, detoxification of xenobiotics, and as sites of liver disease initiation [59]. The Wnt/β-catenin pathway also appears to be required for functional heterogeneity in hepatocytes during the process of hepatic metabolic zoning. Under physiological conditions, this pathway is activated in pericentral hepatocytes. This seems to be due, in part, to the absence of APC in the differentiated pericentral area. Thus, APC is termed a “zone guardian” as negative regulator of the hepatic lobule [60,61,62]. Although the role of APC is not entirely clear, especially in relation to liver parenchyma oxygenation capacity, since hypoxia causes reduced levels of APC mRNA and protein through a direct Hypoxia-Inducible Factor 1-alpha (HIF-1α)-dependent mechanism [62,63]. Moreover, regenerative functions following liver damage could be more or less regulated by the levels of hypoxia and oxidative stress derived from specific mechanisms of tissue damage. The same mechanisms that participate in hepatic pathogenesis may lead to benign and malignant hepatobiliary diseases [62]. All intracellular pathways are inextricably linked to the activity of other pathways, and even β-catenin-independent Wnt signaling could be modulated and, in turn, modulate both the physiological functions of hepatic zoning and regeneration as well as the tumor progression.

## 3. Wnt/β-Catenin Deregulation Signaling in PLC

The loss of functional homeostasis in the hepatic microenvironment also involves the network of signals relying on the Wnt/β-Catenin pathway. Indeed, the dysregulation of this pathway seems to be one of the main drivers of liver carcinogenesis. Thus, the overcoming of the biological capacity, due to the damage, may induce an excessive activation of this pathway that promotes tumor initiation and progression of PLC.

### 3.1. HCC

β-Catenin plays an important function in promoting HCC, based on the frequency of its mutations associated with aberrant Wnt signaling [64], whereas loss of APC function due to mutations is much less frequent. Strong activation of β-catenin signaling was also observed in mouse models with HCC, developed after acute loss of APC in the liver [28]. Although nuclear accumulation of β-catenin appears to be limited to the late stage of the disease, cancer promotion could be due to the driving function of E-cadherin-associated β-catenin on the membrane, which enhances growth factor receptor signaling [44]. The Wnt/β-catenin pathway is frequently upregulated in HCC and is implicated in drug resistance, tumor progression and metastasis [7]. Wnt/β-catenin signaling is activated in zone 3 around the central vein, and this underlines its involvement in liver metabolic zonation [25,65]. Wnt genes (WNT3, WNT4 and WNT5A), and Fz genes (FZD3, FZD6 and FZD7) are upregulated in 60–90% of HCC patients [7].

In human HCC, nuclear β-catenin positivity is observed in 37.7% of patients, with a unicentric nuclear distribution in 53% of cases. The majority of these shows a well-differentiated histology [64]. Patients with β-catenin positivity HCCs develop recurrence and progression of disease. In addition, β-catenin-positive HCC is associated with normal Alpha-fetoprotein (AFP) levels, unicentric tumors with well-differentiated histology and poor prognosis [64]. In HCC, β-catenin expression ranges between 12% and 80% and represents a potential biomarker [7,64]. β-catenin accumulation in nuclei is described in 40–70% of HCCs, while moderate membranous staining is observed in adjacent normal tissue [6,64]. Interestingly, β-catenin mutation may associate with specific radiological features. Indeed, HCC patients show decreased uptake of gadoxetic acid-enhanced magnetic resonance imaging comparing with normal liver tissue [65]. In HCC, β-catenin activation is associated with CTNNB1 mutation in exon 3, by molecular mechanisms that lead to β-catenin stabilization and enhance its nuclear translocation [7]. These activating mutations occur in 18.5% of HCCs [21,66,67]. Potential mechanisms involved in activation of β-catenin in HCC include CTNNB1, AXIN and APC somatic mutations, FZD7 and WNT3 overexpression, SFRP1 and SFRP5 repression [7,68]. Overexpression of genes that play a key role in the development of HCC is crucial in early disease and could be of great value for diagnosis, prognostic prediction or even targeted drug development [69,70].

The presence of CTNNB1 mutations is associated with an increased tumor size, microvascular invasion in HCC [71]. Based on Wnt pathway aberration, Lachenmayer and coworkers suggested a molecular classification of HCC in two groups: CTNNB1 molecular class and Wnt-TGFβ molecular class [72]. The HCC subclass with the CTNNB1 mutation is characterized by upregulation of specific Wnt-targets, low grade and well-differentiated tumors with a favorable prognosis. On the contrary, HCC subclass without the CTNNB1 mutation is characterized by dysregulation of classic Wnt targets and aggressive phenotype [6,73]. In HCC patients, exome sequences revealed cooperating mutations of CTNNB1 with AT-Rich Interaction Domain 2 (ARID2), NFE2 Like BZIP Transcription Factor 2 (NFE2L2), TERT, Apolipoprotein B (APOB) and Lysine Methyltransferase 2D (MLL2). Around the 9–12.5% of human HCC present Met proto-oncogene over-expression/activation and mutations in β-catenin [28]. Exome sequencing has also revealed cooperating mutations of CTNNB1 with albumin (ALB), APOB, Hepatic Nuclear Factors 1 alpha (HNF1A) and 1 alpha (HNF4A). Some observations suggest that the CTNNB1 mutation could be a late event during liver carcinogenesis while accumulation of β-catenin is detected in the early stage of HCC development [6,74]. Wnt/β-catenin signaling is inhibited by the over-expression of miR-300 [75].

In HCC cells, canonical Wnt signaling can be antagonized by non-canonical Wnt5a or even remain simultaneously activated, as occurs via PFTK1 that can interact with cyclin Y (CCNY) [76,77]. In HCC, invasion promotion may occur through the β-catenin/c-Myc pathway as a result of crosstalk between MUC1 and c-Met [78]. Expression of Fz receptors appears over-regulated in more than 60% of HCC cells [79,80]. Accumulated cytoplasmic expression of β-catenin was associated with poor histologic differentiation, increased tumor diameter, and reduced disease-free survival [81,82]. HCCs with CTNNB1 mutations appear morphologically heterogeneous, although nuclear increase of β-catenin in HCCs has been associated with Ki67 expression, reflecting its role in promoting tumor proliferation [74,83]. A link between Wnt/β-catenin signaling and the cell cycle has been suggested. Indeed, CTNNB1 silencing induces modulation in Cyclin B1 and Cyclin C protein expression [84]. Furthermore, several studies in human HCC have observed that β-catenin activation induces cyclin D1 overexpression. However, one must also consider that cyclin D1 is a downstream effector dependent on several molecular cascades in cellular signaling networks, [85]. This would allow β-catenin signaling to cooperate with other molecular drivers in the development and progression of HCC [86,87].

### 3.2. CCA

While HCC is human malignancy that originates from hepatocytes, CCA is a neoplasia that develops from biliary tree cells. CCA is the second most common primary liver cancer after HCC that presents activation of Wnt/β-catenin signaling. CCA development is associated with a state of chronic inflammation sustained by a prolonged exposure of the cholangiocytes to different risk factors, such as infectious agents. In bile ducts this condition leads to activation of different pathways, including Wnt signaling, and promotes cell proliferation, genetic and epigenetic mutations predisposing to CCA [88]. More frequent mutations in CTNNB1 (1.5%), AXIN1 (4%), and APC (2%) genes have been described in this tumor [70]. The host response to infections induces differential expression of Wnt signaling components, in particular upregulation of the non-canonical Wnt5a ligand in infiltrating activated macrophages in inflammation as well as cancer [89]. Macrophages in the stroma of CCA are not resident Kupffer cells, but they derive from adult hematopoietic cells and have an inflammatory origin. Moreover, they express the Wnt7B ligand in the cytoplasm in addition to Wnt7B expression on the surface of the malignant epithelial cells of CCA [90]. Inflammatory macrophages activate the pathway in cholangiocarcinogenesis through production and upregulation of Wnt7b and Wnt10a, that bind on FZD receptors and LRP5/LRP6 co-receptors of cholangiocytes [90]. Moreover, macrophages determine the release of pro-inflammatory cytokines that are inducers of Wnt5a [91]. Tumor-associated macrophages (TAMs) are involved in CCA cell proliferation, angiogenesis and metastasis and are associated with Wnt/β-catenin pathway activation [92]. Indeed, the Wnt/β-catenin pathway plays a role in induction, progression, epithelial-mesenchymal transition (EMT) and multidrug resistance of CCA in association with microRNAs, and other signaling pathways, such as PI3K/AKT/PTEN/GSK-3β [56]. β-catenin show positive staining in 58.3% of CCA samples and a larger tumor size than β-catenin negative cases [5]. Immunohistochemistry (IHC) analysis of surgically resected intrahepatic CCA samples shows β-catenin presence in the cytoplasm and/or nucleus, and its reduction in the plasma membrane. Increased expression of nuclear β-catenin is observed in hilar and extra-hepatic CCA [5]. Increased β-catenin nuclear localization and reduced levels on the plasma membrane correlate with CCA malignancy [93]. Mutation in the β-catenin gene is found in 8.3% of human CCA samples [5]. The mutations of CTNNB1 and AXIN1 also promote CCA cell proliferation and alter apoptotic signaling. In human CCA, IHC staining revealed that Wnt3a, Wnt5a and Wnt7b were positive in 92.1, 76.3 and 100% of case, respectively [94]. Up-regulation of different Wnt ligands, including Wnt2, Wnt3, Wnt5, Wnt7 and Wnt10, has been reported in CCA with changes in β-catenin expression and, in particular, Wnt5a is already upregulated in biliary disease [32,95]. Molecular evidence suggests that while Wnt10 may be involved in the induction of CCA, Wnt7b may promote its progression, likely due to CD68+ macrophage infiltrate that appears to be the main source of Wnt7b in CCA [96,97].

Intrahepatic CCA is associated with cirrhosis caused by HCV, while peripheral CCA is mainly attributed to primary sclerosing cholangitis [98]. Although there are differential risk factors for peripheral and intrahepatic CCA, in both subtypes cytoplasmic β-catenin levels are elevated, whereas its nuclear component appears to be higher in hilar CCA (66.9%) than in intrahepatic CCA (41.9%) [15]. The poor prognosis of intrahepatic CCA could be related to the loss of SFRP1 that induces higher expression of β-Catenin [93].

### 3.3. HB

HB might occur from hepatic stem cells. It shows various histological features (epithelial, mesenchymal, fetal and embryonal) within the same tumor [99]. Based on molecular analysis of β-catenin, two main subtypes can be distinguished: the fetal subtype C1, with a favourable outcome, that shows increased membranous staining and cytoplasmic accumulation of β-catenin, with occasional nuclear localization; subtype C2, which is poorly differentiated and proliferative and characterized by intense nuclear staining of β-catenin [9,19,100]. Oncogenic β-catenin mutations are highly prevalent in HB and paediatric HCC. In-frame somatic mutations of CTNNB1, due to interstitial deletions of exon 3, are prevalent in HB compared to point mutations that are more present in HCC. Furthermore, tumor cells can simultaneously express the mutant and wild-type allele [19]. β-catenin activation is found in up of 87% of human HB and the DNA deletions in exon 3 in β-catenin occur in the majority of sporadic HB. Among human cancers, HB presents the highest rate (up to 90%) of β-catenin mutations [18,19,20,27].

Classical HBs show specific hypomethylated enhancers carrying binding sites for Achaete-Scute Family BHLH Transcription Factor 2 (ASCL2), an essential factor for the gene transcription regulation in the Wnt pathway into definitive endoderm cells [101]. Protracted ASCL2 upregulation, along with fetal-liver-like IGF2 promoter methylation patterns, indicates a probable cell derivation from the premature hepatoblast, similar to the highly proliferative intestinal epithelial cells [101].

Interestingly, β-catenin mutations have no prognostic value due to the similar rates of mutations found in all HB histotypes and at different tumor stages [100]. The severity of HB is directly linked to intracellular accumulation of β-catenin protein from the cell surface to cytoplasm and in the nucleus [19]. A strong nucleocytoplasmatic signal of β-catenin is reported in HB cells metastasized to lungs. Poorly differentiated subtypes of HB exhibit intense nuclear staining of β-catenin. On the contrary, the well-differentiated fetal subtype of HB exhibits membranous staining of β-catenin [102]. The fetal type shows enhanced membranous staining and cytoplasmic accumulation of β-catenin with occasional nuclear localization [19]. In clinical studies, the nuclear co-localization of β-catenin and YAP-1 is identified in approximately 80% of HB cases [18,28,103]. A β-catenin-YAP functional interaction is observed specifically in HB and not in HCC or CCA [18]. The YAP signaling pathway cooperates with β-catenin in HB pathogenesis but the molecular mechanisms that contribute to development and growth of HB remain to be discovered [20]. Recurrent mutations in CTNNB1 and increased expression of nuclear factor erythroid 2-related factor2 gene (NFE2L2), that is involved in the response to oxidative stress, have been reported in HB [18,20]. In particular, the NFE2L2/NRF2 pathway is involved in the control of the innate immune response, cytosolic DNA sensing in viral infection, and survival in sepsis. Moreover, it regulates the redox equilibrium and prevents the dysregulation of MyD88-dependent or independent, or TNF-alpha proinflammatory signaling. In response to oxidative stress, NFE2L2/NRF2 induces the transcription of different cytoprotective genes by binding to antioxidant response element (ARE) of their promoters [103,104,105].

## 4. Conventional Therapeutic Strategies in PLC

Surgical resection is the treatment of choice for PLC patients with well-preserved liver function. The overall 5-year survival rate of patients with advanced PLC is around 5-8.9% [3,64]. Specifically, survival of advanced HCC is approximately eight months without any intervention, but in expert oncological Centers the HCC prognosis is extended beyond four years [106,107]. In the surgical approach for HCC and CCA, negative resection margins can be achieved in less than 30% of patients, especially in CCA [4,107,108]. Indeed, CCA is a fatal malignant tumor with the ability to metastasize in 60–70% of patients with poor prognosis due to its late diagnosis at the advanced stage [4,97]. Overall survival with the current available chemotherapy regimen based on gemcitabine and cisplatin, is less than one year [109]. Among diseases across the pediatric population, HB is the most common pediatric tumor with an incidence of 1.2–1.5 million children per year. HB is of different subtypes according to histological features [102]. For low-risk HB the survival rates are more than 90%, while in cases of high-risk category the overall survival is only 25–40% [20].

PLCs are very difficult to treat. Systemic chemotherapy, with or without the association of immunotherapy, has significant side effects in PLC patients. Several additional combined therapies are currently being tested in clinical trials but there is an urgent need to identify novel potential molecules with more individual therapeutic timing. For this, all efforts must made in developing direct-acting treatment modalities in selected area of the liver. Future target therapies have to guarantee a selective action in the PLC nodule, with the aim of saving normal liver parenchyma.

### 4.1. HCC Treatments

Activation of Wnt/β-catenin signaling may induce two different phenotypes of HCC: the “Hyde phenotype” defined by cancer stem cell feature, invasion and metastasis, and the “Jekyll phenotype” defined as the HCC subset with good prognosis [73]. Treatment decisions for HCC patients are complex and depend on tumor staging, portal hypertension, degree of liver dysfunction and medical expertise [107]. Hepatectomy is curative when HCC is confined to the liver, while liver transplantation can be curative in unresectable disease [107]. These treatments are considered curative in 30% of patients with early-stage HCC (Barcelona Clinic Liver Cancer, BCLC stage 0 or A) [110]. Currently, alternative approaches for HCC patients, that are not eligible for surgery, include radio or chemo-embolization, microwave ablation and chemotherapy with molecular target therapies [6,7]. Transarterial chemoembolization (TACE) or transarterial bead embolization (TABE) are indicated for HCC patients with BCLC B [106]. Although of great interest, the combination of TACE and Sorafenib does not appear additive, and the data obtained from this protocol are inconclusive [106]. Transarterial radioembolization (TARE) is indicated in unresectable HCC patients, which is a form of brachytherapy. TACE and TARE show similar performance with reduced toxicity in TARE procedures [106]. TACE is contraindicated in HCC cases with portal vein thrombosis, while radiofrequency and microwave ablations are contraindicated in HCC patients with bleeding diathesis [110].

Sorafenib is an anti-angiogenic and MAP kinase inhibitor active against HCC that improves survival in patients with advanced disease [6,111]. In 2007, Sorafenib was approved by the US Food and Drug Administration (FDA) as the first line treatment for advanced HCC [110]. However, the median overall survival of patients treated with Sorafenib was prolonged by only 2.8 months compared with placebo-controlled group [7,110]. On the other hand, some studies document an increased overall survival, from 8 to 11 months, in HCC patients with extrahepatic spread after Sorafenib treatment. This drug has also adverse events, including diarrhea, hand-foot skin reaction, weight-loss and hypophosphatemia [110]. In 2019, atezolizumab in combination with bevacizumab were approved as first-line therapy for patients with advanced HCC [111]. More recently, molecularly targeted therapies with lenvatinib and regorafenib, cabozantinib and ramucirumab have been approved due to the survival advantages documented in phase III clinical trials [112]

### 4.2. CCA Treatments

CCA treatments include radical surgery, chemotherapy, radiotherapy and their combination [4]. Surgery is the preferred treatment modality for CCA [95]. Liver transplantation is an option for small CCA lesions with intrahepatic growth. The current standard first-line treatment for CCA non-suitable for surgery or loco-regional therapies is the combination of gemcitabine and cisplatin [4]. The use of second-line therapy depends on the performance status of CCA patients. The treatment schedules are based on fluoropyrimidine, irinotecan, docetaxel, gemcitabine and platinum-compounds [4].

### 4.3. HB Treatments

Current primary treatments for HBs include curative surgical resection and cisplatin-based chemotherapy, while liver transplantation represents the recommended therapeutic option in advanced stage tumors with vascular infiltration [18]. In particular, use of cisplatin contributes to improve the 5-year-survival rate to over 70% according to different studies [113,114]. Although the International Childhood Liver Tumor Strategy Group (SIOPEL) currently suggests starting preoperative cisplatin based-chemotherapy and deferring surgical resection after 2–3 months of therapy, in most patients high-dose drug treatment induces hearing loss linked to its ototoxicity [113,114]. Current treatments are ineffective in 20–30% of HB cases and, unfortunately for these patients, overall survival remains poor [20,113].

## 5. The Wnt/β-Catenin Signaling Pathway as a Target of Anti-PLC Drugs

Therapeutic approaches targeting Wnt/β-catenin signaling consist of antibodies against Wnt ligands and Fz receptors, or molecules that inhibit their secretion and interaction. Other options include inhibition of palmitoylation and secretion of Wnt ligands. Regarding β-catenin, some molecules have been developed to promote its degradation at the cytoplasmic level, or the inhibition of its nuclear transduction [6]. Currently no approved therapeutic agents are available for clinical use that can target the Wnt/β-catenin pathway for PLC (see Table 1). Chemotherapy and radiation can upregulate Wnt signaling, and this event protects cancer cells from cell cycle arrest or apoptosis. In other terms Wnt signaling mediates resistance to conventional chemotherapies [115]. Therefore, several compounds have been developed to target the Wnt/β-catenin pathway. They are schematically divided into inhibitors of Wnt ligands using porcupine molecules (PORCN), molecules that interact with Wnt receptors and coreceptors (Frizzled and LRP5/LRP6) [116], molecules that interact with cytoplasmic signaling (tankyrase and CK1alpha), and molecules that interact with intranuclear Wnt signaling. In addition to these molecules, other approved drugs commonly used in clinical practice have been shown to possess activity against the Wnt cascade but, unfortunately, their antitumor potency has not been established in clinical use. They include indomethacin, pyrvinium, sulindac, aspirin, celecoxib, and rofecoxid [87].

Small molecules inhibiting PORCN have been developed to prevent palmitoylation of Wnt proteins and their secretion, and they can shut down both canonical and non-canonical Wnt signaling [115,117]. PORCN inhibitors such as LGK-974 and ETC-159 might suppress the Wnt cascade [87]. Currently, a growing number of reports show the side effects of PORCN inhibitors, such as bone loss caused by diminished osteogenesis and increased osteolysis [118]. CGX1321 is a novel PORCN inhibitor used in solid tumors in clinical trials and has been tested in HCC and CCA patients in a phase I trial (see Table 1) [118,119].

Fz receptors have a significant role in cancer progression and may contribute to the therapeutic target of PLC. Monoclonal antibodies could increase apoptosis and prevent cell proliferation by inhibiting Wnt ligands and Fz receptors. Because Fz receptors are up-regulated in HCC, they are a good molecular target. Based on these considerations, it has been reported that soluble Fz7 (sFz7) can also compete with Fz receptors for the Wnt3 ligand. Vantictumab (OMP-18R5) is a novel monoclonal antibody that interacts with these Fz receptors to block canonical Wnt signaling [68]. It has been evaluated in phase I trials combined with Fz7, Fz1, Fz2, Fz5 and Fz8. To date, there are two ongoing clinical phase I/II trials using OMP-18R5, modulating the ligand/Fz receptor interfaces, and PRI-724, interfering β-catenin dependent gene transcription [7]. A phase I trial is testing the value of the Fz8-Fc fusion protein Ipafricept (OMP-54F28) in HCC patients [120]. OMP-54F28 is a recombinant protein that competes for binding with the Fz8 receptor by sequestering Wnt ligands [117,118]. This molecule shows a high antitumor effect in association with gemcitabine and taxanes. Moreover, it was evaluated in patients with advanced HCC, in a phase I single group trial (see Table 1), in combination with Sorafenib [119].

In HB treatment, Fz receptors may serve as useful therapeutic target by gene expression inhibition [121].

Dickkopf-1 (DKK1) is a secreted modulator of Wnt signaling, but it shows immunosuppressive effects in many cancers when overexpressed and is often associated with worse clinical outcomes. DKN-01 is a humanized IgG4 antibody used to block the activity of DKK1, enhancing innate immune responses by the immunosuppressive effects of DKK1 in the tumor microenvironment [122]. Most clinical trials have been designed to study the effects on DKN-01 in gastroesophageal, intestine, liver and biliary tract cancers [118]. The efficacy of humanized DKN-01 was evaluated in a phase I trial in HCC and CCA patients in combination with gemcitabine and cisplatin [123,124]. This therapeutic approach revealed that more than 30% of patients experienced a partial response in advanced biliary cancer [68]. A phase II trial (NCT03645980) studied the effect of DKN-01 in patients affected by advanced HCC using the combination with Sorafenib (Status: recruiting). Eads 2016 (NCT02375880) is a single-group phase I study investigating the effect of DKN-01, in patients with advanced CCA, in combination with gemcitabine and cisplatin (Status: ongoing) [125].

Tankyrase inhibitors (TNKSi) cause stabilization of Axin 1/2 proteins, β-catenin degradasome accumulation and blockade of WNT/β-catenin signaling [125]. TNKsi inhibitors include XAV939, G007-LK, G244-LM, RK-287107, JW55, K-756, IWR-1, MSC2504877, AZ1366, JW74 and NVP-TNKS656 [87]. Tankyrase inhibitor NVP-TNKS656 has shown tumor-suppressive effects in HCC preclinical models but has not entered into clinical trials [121,126].

β-catenin signaling increases the metastatic capacity of HCC cells through upregulation of pyruvate dehydrogenase kinase isoenzyme-1 (PDK1) to stimulate the Warburg effect and energy delivery to the tumor. In contrast, PDK1 activation can be suppressed by PPARγ-coactivator-1α (PGC-1α), although PPARγ appears to be upregulated in HCC cells and human HCC tissues resistant to Sorafenib [127]. FH535, a compound that suppresses both Wnt/β-catenin and PPAR signaling, has also been investigated in combination with Sorafenib in liver cancer stem cells (CSCs) and advanced HCC [7,128]. This approach opened an alternative for heterogeneous disease treatments by targeting different molecular pathways. The presence of CSCs contributes to metastasis, recurrence, and resistance to chemo/radiotherapy in HCC. The Wnt signaling pathway is related to stemness maintenance. IC-2 is a novel small-molecule Wnt inhibitor that reduces the CD44+ cell population (CSCs) in HCC [117,129]. In addition to the development of new antagonistic molecules, pharmacological research for the inhibition of Wnt/β-catenin signaling in PLC could be directed towards the repurposing of non-oncology drugs, already active for other diseases, and the evaluation of natural compounds that may have an anti-inflammatory effect on the tumor microenvironment [130] (see Table 2). Such drugs may also be evaluated individually or in combination with each other. Indeed, some trials have been designed to study the effects of Hydroxy-chloroquine in unresectable HCC plus transarterial chemoembolization or plus Sorafenib, as well as in advanced CCA in association with Opaganib (ABC294640) [68]. In addition, Molenaar 2017 (NCT02496741), in a phase II single group trial that studied the effect of v-ATPase inhibitor in intrahepatic CCA, used a combination of Chloroquine and Metformin (Status: completed) [131].

Concerning HB treatment, there are only preclinical studies that prospect a target therapy for Wnt/β-catenin signaling, the use of small molecules in combination with selective COX2 inhibitors, or natural compounds and repurposing of drugs showing anticancer effects, such as epigallocatechin-3-gallate [102,132,133,134,135,136]. ICG-001 is a novel small-molecule inhibitor of Wnt signaling that disrupts β-catenin-CREB binding [137]. It reduced HB cell viability in the presence of celecoxib and enhanced the anti-tumor activity of sorafenib in an HB animal model [138,139].

## 6. Conclusions

The Wnt/β-catenin signaling pathway physiologically regulates cell proliferation and differentiation in the liver parenchyma through a complex network of interactions modulated by the functional requirements of the microenvironment. These interactions are lost during cancerogenesis and tumor progression in which the Wnt/β-Catenin signaling is dysregulated.

Currently, applied therapeutic protocol tend to fight human liver cancer, although it should be possible to act more selectively and save the normal liver parenchyma. To this end, it is necessary to better understand the functions of the Wnt/β-catenin pathway in physiological function and liver regeneration.

Combinations of drugs, using conventional and experimental molecules, should be tested and administered using selective intrahepatic transarterial procedures rather than systemic chemotherapeutic approaches. The selection of liver parenchyma might be of great interest due to the importance of reducing the toxicity of Wnt inhibitors in normal liver tissue and increasing the inhibitory effects specifically on hepatic cancer cells. Novel therapeutic drug delivery nanotechnologies associated with directional regulation of Wnt signaling could delineate new perspectives and personalized treatment opportunities for PLC care that are urgently needed in clinical practice.

## Figures and Tables

**Figure 1 cancers-14-01912-f001:**
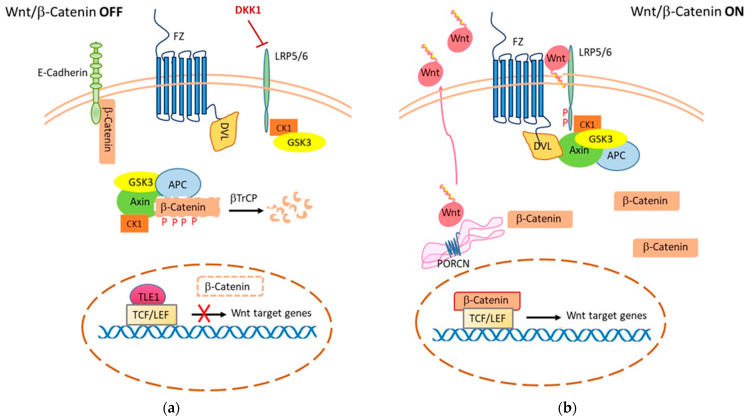
Wnt/β-Catenin-dependent signaling. (**a**) In the OFF state, or absence of Wnt ligand, cytoplasmic β-catenin undergoes phosphorylation by the destruction complex (composed of Axin, APC, CK-1, and GSK-3). This leads to degradation of β-catenin in the proteosome after being ubiquitinated by β-TrCP. In the nucleus, transcription of target genes is repressed through the TLE-1 complex with TCF/LEF. (**b**) In the ON state, porcine acylase (PORCN) o-acylates Wnt ligands and facilitates their secretion and interaction with receptors. Binding of Wnt to the Frizzled receptor causes phosphorylation of the LRP co-receptor by CK1 and GSK3. Dvl protein is recruited to the plasma membrane and blocks the β-catenin destruction complex. This causes β-catenin to accumulate in the cytoplasm and then translocate to the nucleus, where it forms a complex with TCF/LEF to transcribe its target genes (i.e., cyclin D1, c-Myc, vascular endothelial growth factor (VEGF), interleukin-8 (IL-8), etc.).

**Figure 2 cancers-14-01912-f002:**
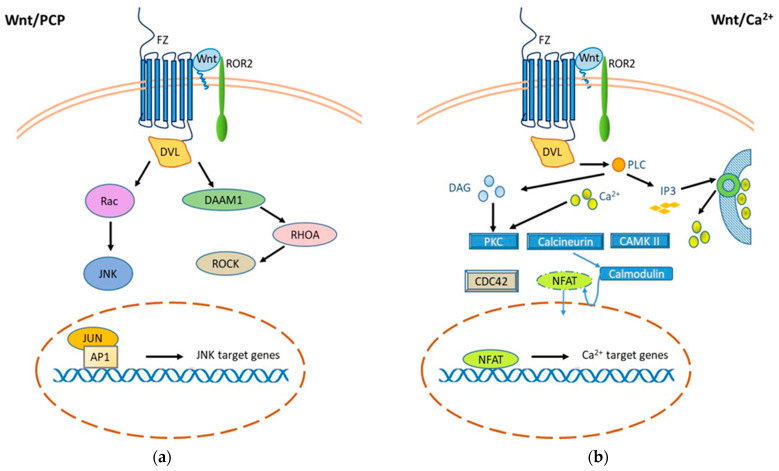
Wnt/β-Catenin-independent signaling. Two non-canonical β-catenin-independent Wnt signaling pathways have been reported, both of which are activated by binding of a Wnt ligand to Fz proteins and their co-receptor tyrosine kinase-like orphan receptor 2 (ROR2) to activate Dishevelled (Dvl). (**a**) The planar cell polarity (PCP) pathway mediates cell adhesion, migration, and cytoskeleton organization. Activation of a trimeric G protein induces activation of Dvl and DAAM, which downstream activate the small GTPases RHOA and RAC-1 and the kinases JUN (JNK) and ROCK. The signal is amplified by the activation of JNK, which phosphorylates the transcription factors AP-1 and JUN that translocate into the nucleus and regulate gene expression. (**b**) The Wnt/Ca^2+^ signaling pathway is mediated by G proteins with activation of phospholipase C, leading to hydrolysis of PIP2 to DAG and IP3. IP3 induces the diffusion of Ca^2+^ ions from within the compartments in which they are stored into the cytoplasm to activate calcineurin and CAMKII. Intracellular calcium and DAG activate protein kinase C (PKC) by increasing calcineurin activity, which causes translocation of the transcriptional factor NFAT into the nucleus and target genes transcription.

**Table 1 cancers-14-01912-t001:** Wnt/β-catenin inhibitors in clinical and pre-clinical trials for human HCC and CCA.

Drug	Target	Cancer Type	Phase	Clinical Trials	References
CGX1321	PORCN	HCC CCA	I	NCT03507998 *	[117] [112]
DKN-01	DKK1	HCC CCA	I/II I (in combination with Gencitabine and Cisplatin)	NCT03645980 ^§^ NCT02375880	[112] [118]
OMP-54F28 + Sorafenib	FZD8	HCC	I	NCT02069145 ^#^	[119]
Salinomycin	LRP5/6 inhibitor	HCC	Pre-clinical		[116]
NVP-TNKS656	Tankyrase inhibitor	HCC	Pre-clinical		[120]
IC-2	Wnt	HCC	Pre-clinical		[121]

* Curegenix Inc. & Merck, S. and D.C. Phase I Dose-Escalation Study of CGX-1321 in Subjects with Advanced Gastrointestinal Tumors (NCT03507998) (2018). Available online at: http://clinicaltrials.gov/show/NCT03507998 ^#^ (accessed on 15 February 2022). OncoMed Pharmaceuticals, Inc.(Redwood City, CA, USA)A Phase 1b Dose Escalation Study of OMP-54F28 in Combination with Sorafenib in Patients with Hepatocellular Cancer (2014). Available online at: https://clinicaltrials.gov/ct2/show/NCT02069145. 54F28-004 ^§^ (accessed on 15 February 2022). Marquardt, D. J. U. A Phase I/II Multicenter, Open-label Study of DKN-01 to investigate the anti-tumor activity and safety of DKN-01 in patients with hepatocellular carcinoma and Wnt signaling alterations. Available online at: https://clinicaltrials.gov/ct2/show/NCT03645980 (accessed on 15 February 2022).

**Table 2 cancers-14-01912-t002:** Up-to-date drugs that have implications on Wnt/β-catenin signaling in HCC.

Drug	Possible Mechanisms on Wnt Signal	Cells Models	References
Shizukaol D Chloranthus serratus	Unknown	SMMC-7721 SK-HEP1 HepG2	[116] [116] [116]
Curcumin	Inhibiting GPC3/TPA-induced Wnt signal activation	HepG2 Hep3B	[133] [133]
Pimozide	Unknown	Hep3B HepG2	[135] [135]
Ethacrynic acid	Unknown	Hep3B HepG2	[136] [136]
Epigallocatechin-3-gallate	Inhibition of Wnt signaling (decreasing c-myc expression and causing the induction of SFRP1) in HB		[102]
ICG-001	Disruption β-catenin-CREB binding in HB	HuH6 HepT1	[138]
Salinomycin FH-535	Increasing intracellular Ca^2+^ levels Inhibition of β-catenin signaling and peroxisome proliferator-activated receptor (PPAR) in HCC	HepG2 BEL-7402	[116] [116] [128]

In vitro studies prospecting a combination target therapy for Wnt/β-catenin signaling using small molecules in combination with selective inhibitors, natural compounds and repurposed drugs, showing anticancer effects.

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
