# Peer review of "Targeting Wnt/β-Catenin Pathways in Primary Liver Tumours: From Microenvironment Signaling to Therapeutic Agents"

_cancers, 2022, doi:10.3390/cancers14081912_

Round 1
Reviewer 1 Report
Selvaggi et al presented a review article about the therapeutic targeting of wnt-b-catenin signaling in liver tumors. The topic of the manuscript while of potential interest, has however been recently (last 2 years) reviewed before in
- Nat Rev Gastroenterol Hepatol. 2019 Feb;16(2):121-136. doi: 10.1038/s41575-018-0075-9. Wnt-β-catenin signalling in liver development, health and disease
- Biomed Pharmacother. 2020 Dec;132:110851. doi: 10.1016/j.biopha.2020.110851. Epub 2020 Oct 17. WNT/β-catenin signaling in the development of liver cancers
- Cancer Biol Med. 2021 Oct 1;j.issn.2095-3941.2021.0306. doi: 10.20892/j.issn.2095-3941.2021.0306. Online ahead of print. The Wnt/β-catenin signaling pathway in the tumor microenvironment of hepatocellular carcinoma
- Cancers (Basel). 2019 Jul 2;11(7):926. doi: 10.3390/cancers11070926. Wnt/β-Catenin Signaling in Liver Cancers
- J Hepatocell Carcinoma. 2021 Nov 25;8:1415-1444. doi: 10.2147/JHC.S336858. eCollection 2021. Wnt/β-Catenin Signaling as a Driver of Hepatocellular Carcinoma Progression: An Emphasis on Molecular Pathways
- Medicine in Drug Discovery. Volume 13, March 2022, 100113 Wnt/β-catenin signaling pathway in lung cancer
- J Clin Invest. 2022 Feb 15;132(4):e154515. doi: 10.1172/JCI154515. β-Catenin signaling in hepatocellular carcinoma
- Cancers (Basel). 2021 Apr 12;13(8):1830. doi: 10.3390/cancers13081830. β-Catenin Activation in Hepatocellular Cancer: Implications in Biology and Therapy
Moreover, this review manuscript has serious flaws that may impede its publication in Cancers, since in some parts of the manuscript I detected plagiarism. Moreover, the manuscript presents descriptive sentences and references mostly focused in other review articles and not original recent and relevant original research reports. Some specific drawbacks for example:
Examples:
Lines 92-93, 98-99 lack a reference.
line 345 - reference 7 does not report information about miR-300
Lines 346-349 are ipsis verbis a copy of text from reference 3
Lines 350 – 353 are ipsis verbis a copy of text from reference 7
Author Response
Authors' responses to 1st reviewer
We thank the reviewer for his careful reading of the manuscript and his constructive remarks to improve the manuscript. We have taken inspiration from the comments to improve and enhance the unique features of our manuscript. Please find below a detailed point-by-point response to all comments (reviewers’ comments in black, our replies in blue).
- Selvaggi et al presented a review article about the therapeutic targeting of wnt-b-catenin signaling in liver tumors. The topic of the manuscript while of potential interest, has however been recently (last 2 years) reviewed before in
- Nat Rev Gastroenterol Hepatol. 2019 Feb;16(2):121-136. doi: 10.1038/s41575-018-0075-9. Wnt-β-catenin signalling in liver development, health and disease
- Biomed Pharmacother. 2020 Dec;132:110851. doi: 10.1016/j.biopha.2020.110851. Epub 2020 Oct 17. WNT/β-catenin signaling in the development of liver cancers
- Cancer Biol Med. 2021 Oct 1;j.issn.2095-3941.2021.0306. doi: 10.20892/j.issn.2095-3941.2021.0306. Online ahead of print. The Wnt/β-catenin signaling pathway in the tumor microenvironment of hepatocellular carcinoma
- Cancers (Basel). 2019 Jul 2;11(7):926. doi: 10.3390/cancers11070926. Wnt/β-Catenin Signaling in Liver Cancers
- J Hepatocell Carcinoma. 2021 Nov 25;8:1415-1444. doi: 10.2147/JHC.S336858. eCollection 2021. Wnt/β-Catenin Signaling as a Driver of Hepatocellular Carcinoma Progression: An Emphasis on Molecular Pathways
- Medicine in Drug Discovery. Volume 13, March 2022, 100113 Wnt/β-catenin signaling pathway in lung cancer
- J Clin Invest. 2022 Feb 15;132(4):e154515. doi: 10.1172/JCI154515. β-Catenin signaling in hepatocellular carcinoma
- Cancers (Basel). 2021 Apr 12;13(8):1830. doi: 10.3390/cancers13081830. β-Catenin Activation in Hepatocellular Cancer: Implications in Biology and Therapy
Authors' responses:
- In writing this manuscript, we aimed to provide a comprehensive view of the role played by Wnt/b-Catenin signaling in healthy human liver and in primary human liver tumors. We sought to provide the reader with what we felt was missing in the recent landscape of review articles on Wnt/b-Catenin.
We considered specifically the altered functions of Wnt/b-catenin signaling in human primary liver cancers, with emphasis on hepatocellular carcinomas (HCC), cholangiocarcinomas (CCA) and hepatoblastomas (HB).
Based on recent literature, we focused on liver carcinogenesis through deregulation of Wnt signals.
We tried to provide a comprehensive view on the topic, describing in one article very complex concepts, from the physiology of the tissue microenvironment to tumor pathology through molecular deregulation of Wnt/b-Catenin. Finally, we described the clinical implication of Wnt signaling inhibitors in combination with conventional and loco-regional surgical therapies, also providing further perspectives for personalized combination therapy.
- Moreover, this review manuscript has serious flaws that may impede its publication in Cancers, since in some parts of the manuscript I detected plagiarism. Moreover, the manuscript presents descriptive sentences and references mostly focused in other review articles and not original recent and relevant original research reports. Some specific drawbacks for example:
Examples:
Lines 92-93, 98-99 lack a reference.
line 345 - reference 7 does not report information about miR-300
Lines 346-349 are ipsis verbis a copy of text from reference 3
Lines 350 – 353 are ipsis verbis a copy of text from reference 7
Authors' responses:
- We apologize for the inconvenience of the overlaps, which were beyond our intent.
We have carefully rechecked the entire manuscript and included multiple original bibliographic sources appropriate to the concepts expressed.
We reworded sentences that showed weaknesses or overlap with other texts.

Reviewer 2 Report
This review is well written and the references are appropriate. For my opinion the paper may be accepted for the pubblication in this journal
Author Response
2° Review Report
This review is well written and the references are appropriate. For my opinion the paper may be accepted for the pubblication in this journal
AUTORS RESPONSES
We thank the reviewer for careful reading of the manuscript and for appreciating our research.
We considered the reviewer's comments to improve and correct the manuscript.
We carefully checked the whole text to make the necessary corrections (shown in the text in blue color) and properly comprehend the manuscript.
Reviewer 3 Report
The manuscript titled “Targeting Wnt/β-catenin pathways in primary liver tumours: from microenvironment signaling to therapeutic agents” reviewed the function of Wnt/β−catenin signaling pathways in liver-related carcinoma. And further perspective the potential combination therapy strategies and possible personal therapeutics in the future clinical applications. Overall, the manuscript is well-written and prepared for the most part. I only have a few marks as the following that the authors may want to consider.
Concerns and Comments:
- Line 40: I’d suggest the authors switch the keyword “therapy” to another one due to it is too broad.
- Line 161: The left half “(“ is not bold, the right half “)” is bold.
- Figure 1: Please use “Wnt” instead of “WNT” in the image. This should be homogenous throughout the whole manuscript
- Figure 2: In the image, the “Ca+2” should be “Ca2+”.
- Please provide the software or online tools that support generating the signaling images.
- Line 229: “Ca+” should be “Ca2+”. Superscript “2+”, “Ca2+”. The same as line 187 and line 229. Please check Table 2 as well.
- Lines 254-292: This paragraph is too long to track. Please consider splitting it into two or three paragraphs to make it clearer.
- There are extra spaces throughout the whole manuscript. Please check. For example, line 354, line 361, line525 after “Wnt/”.
- Lines 470-471: Please cite this paper into serial reference, “[Gutierrez JA et al Trans Cancer Res 2013]”.
- Line 594: The “Wnt/b” should be “Wnt/β”. The same as line 611. Please check.
- Table 1: I’d suggest the authors use “&” to replace “°”, or other preferred symbols for easier tracking.
- I can’t find where Table 2 has been cited or mentioned in the main context. Please check and update.
- The format should be taken care of. The first line indents are not homogenous throughout the manuscript, for example, line 74, line 86.
Author Response
Authors' responses to 3rd reviewer
We thank the reviewer for his careful reading of the manuscript and his constructive remarks to improve the manuscript.
Please find below a detailed point-by-point response to all comments.
3° Review Report
The manuscript titled “Targeting Wnt/β-catenin pathways in primary liver tumours: from microenvironment signaling to therapeutic agents” reviewed the function of Wnt/β−catenin signaling pathways in liver-related carcinoma. And further perspective the potential combination therapy strategies and possible personal therapeutics in the future clinical applications. Overall, the manuscript is well-written and prepared for the most part. I only have a few marks as the following that the authors may want to consider.
Authors' responses to reviewer concerns and comments point by point:
- Line 40: I’d suggest the authors switch the keyword “therapy” to another one due to it is too broad.
We changed the keyword "therapy" to two keywords "surgical treatments" and "chemotherapy."
- Line 161: The left half “(“ is not bold, the right half “)” is bold.
We made uniform the fonts that were left bold in the draft work.
- Figure 1: Please use “Wnt” instead of “WNT” in the image. This should be homogenous throughout the whole manuscript
Figure 2: In the image, the “Ca+2” should be “Ca2+”.
We have rechecked and corrected the inscriptions in the figures
- Please provide the software or online tools that support generating the signaling images.
The images were designed by us using the Microsoft Office PowerPoint.
- Line 229: “Ca+” should be “Ca2+”. Superscript “2+”, “Ca2+”. The same as line 187 and line 229. Please check Table 2 as well.
We have rechecked and corrected Table 2
- Lines 254-292: This paragraph is too long to track. Please consider splitting it into two or three paragraphs to make it clearer.
We have divided paragraph 2 into three subparagraphs to make it clearer.
- There are extra spaces throughout the whole manuscript. Please check. For example, line 354, line 361, line525 after “Wnt/”.
We double-checked the alignment and eliminated the excess spaces in the entire manuscript as well as the form already set up of the journal allowed us.
- Lines 470-471: Please cite this paper into serial reference, “[Gutierrez JA et al Trans Cancer Res 2013]”.
We carefully rechecked the entire manuscript and included multiple original bibliographic sources appropriate to the concepts expressed.
- Line 594: The “Wnt/b” should be “Wnt/β”. The same as line 611. Please check.
We have corrected the letters in symbol font
- Table 1: I’d suggest the authors use “&” to replace “°”, or other preferred symbols for easier tracking.
I can’t find where Table 2 has been cited or mentioned in the main context. Please check and update.
We have double-checked the tables and entered what you requested
- The format should be taken care of. The first line indents are not homogenous throughout the manuscript, for example, line 74, line 86.
We double-checked the format
Round 2
Reviewer 1 Report
Authors accepted some of the proposed suggestions and improved the quality of the manuscript. I now do not have ethical concerns regarding this manuscript.